# A Community-Based Management of COVID-19 in a Mobile Container Unit

**DOI:** 10.3390/vaccines9111362

**Published:** 2021-11-19

**Authors:** Elena Petrova, Timothy Farinholt, Tejas P. Joshi, Hannah Moreno, Mayar Al Mohajer, Shital M. Patel, Joseph Petrosino, Sharmila Anandasabapathy

**Affiliations:** 1Baylor Global Health, Baylor College of Medicine, Houston, TX 77030, USA; tejas.joshi@bcm.edu; 2Department of Medicine—Gastroenterology, Baylor College of Medicine, Houston, TX 77030, USA; 3Alkek Center for Metagenomics and Microbiome Research, Molecular Virology and Microbiology, Baylor College of Medicine, Houston, TX 77030, USA; timothy.farinholt@bcm.edu (T.F.); hannah.moreno@bcm.edu (H.M.); jpetrosi@bcm.edu (J.P.); 4Department of Medicine—Infectious Disease, Baylor College of Medicine, Houston, TX 77030, USA; mayar.almohajer@bcm.edu (M.A.M.); shitalp@bcm.edu (S.M.P.)

**Keywords:** mobile clinic, relocatable medical facility, underserved community, coronavirus, vaccination, infection control, decontamination

## Abstract

Vaccine uptake is a multifactor measure of successful immunization outcomes that includes access to healthcare and vaccine hesitancy for both healthcare workers and communities. The present coronavirus disease (COVID-19) pandemic has highlighted the need for novel strategies to expand vaccine coverage in underserved regions. Mobile clinics hold the promise of ameliorating such inequities, although there is a paucity of studies that validate environmental infection in such facilities. Here, we describe community-based management of COVID-19 through a Smart Pod mobile clinic deployed in an underserved community area in the United States (Aldine, Harris County, TX, USA). In particular, we validate infection control and biological decontamination of the Smart Pod by testing surfaces and the air-filtration system for the COVID-19 virus and bacterial pathogens. We show the Smart Pod to be efficacious in providing a safe clinical environment for vaccine delivery. Moreover, in the Smart Pod, up-to-date education of community healthcare workers was provided to reduce vaccine hesitancy and improve COVID-19 vaccine uptake. The proposed solution has the potential to augment existing hospital capacity and combat the COVID-19 pandemic locally and globally.

## 1. Introduction

Vaccination programs and their overall success result from high effectiveness and high uptake of vaccines [1,2,3]. The limited access to health care of vulnerable, underserved populations, vaccine hesitance, and lack of up-to-date training programs for healthcare workers in community-based clinics lead to reduced vaccine uptake that diminishes the effectiveness of vaccination programs [4,5,6,7]. During outbreaks of highly infectious diseases, such as the current coronavirus disease (COVID-19) pandemic, innovative approaches to cost-effective community-based solutions for vaccine delivery are particularly in demand. According to the most recent report of Centers for Disease Control and Prevention (CDC) on source and accessibility of healthcare in the United States [8], about 17% of patients do not have a regular place to access healthcare services.

Vaccine hesitancy constitutes a major hindrance to vaccine delivery. Of note, studies have shown vaccine hesitancy to be especially pronounced in underserved communities, in part due to the logistical inconveniences of obtaining vaccines [9,10,11,12]. A 2020 questionnaire distributed to 3865 individuals showed economic hardship to be associated with vaccine refusal [11]; additionally, a study of 10,871 participants showed vaccine hesitancy to be fivefold and twofold higher among Black and Hispanic populations, respectively, compared to their White counterparts [12]. Moreover, a recent systematic review of factors contributing to vaccine hesitancy noted that inconvenience and a lack of accessibility to vaccines may be the common denominators in racial and economic disparities in vaccine delivery.

Portable clinics hold the promise of being able to provide high-quality healthcare services to at-risk populations and bridge existing inequities in health, many of which are exacerbated in pandemic and disaster situations. A growing body of research supports that mobile health clinics are a valuable and cost-effective model to healthcare delivery for a variety of clinical applications [13,14,15,16] including vaccination [17,18]. The main strength of mobile clinic solutions is the ability to provide adequate, quick and quality healthcare services for people in locations removed from stationary health care facilities or those that have no access to conventional healthcare services due to social or financial reasons. Though, inconvenient infrastructure, varying location, and unclear policy in some healthcare systems can be among the factors that reduce mobile clinic effectiveness and should be overcome to reduce barriers that hinder widespread implementation of portable health clinics.

Thus, mobile medical units are economically and logistically viable platforms for healthcare delivery that can augment stationary healthcare facilities [15,16] and temporary mass-vaccination centers [3]. During the COVID-19 pandemic, the burgeoning need for testing, vaccine delivery, and management of SARS-CoV-2-induced respiratory disease has highlighted the utility of relocatable clinics with enhanced infection control. For instance, the mobile clinic paradigm has been applied to manage the COVID-19 crisis in a multitude of settings: it has been deployed to provide primary care and COVID-19 testing to migrant farm workers in Iowa [19], monoclonal antibody infusion to patients in nursing homes and long-term care facilities [20], and COVID-19 testing in a disadvantaged Baltimore neighborhood [21].

While environmental contamination control is a part of prevention of nosocomial infection and an established practice for stationary hospitals [22,23], there is a paucity of studies that validate the infection control and biological decontamination measures taken in mobile container medical units dealing with highly contagious pathogens such as SARS-CoV-2.

As nosocomial spread represents an important mode of transmission for pandemic viruses such as SARS-CoV-2, designing and validating a portable clinic from an environmental infection control standpoint are paramount. Furthermore, recent case reports and computer models have supported the possible spread of SARS-CoV-2 viral particles via heating, ventilation, and air conditioning (HVAC) systems, emphasizing the need to validate HVAC systems in mobile clinics for absence of SARS-CoV-2 [24,25].

To deliver high-quality community-based healthcare services and COVID-19 vaccine administration to underserved populations, the portable COVID-19 Smart Pod isolation clinic with advanced air-purification systems, antimicrobial surface features, and a negative pressure isolation room was deployed in Aldine (Harris County, TX, USA) during the COVID-19 pandemic in September 2020 [26]. This rapidly deployable, expandable shipping container-based medical facility serves the dual purpose of (a) everyday community-based care and (b) emergency mobilization during an epidemic or natural disaster [27]. Baylor College of Medicine (BCM) developed and patented the first Smart Pod hospital ward in response to the West African Ebola epidemic (2015) and deployed it in Monrovia, Liberia [28,29,30]. In 2019, the Smart Pod pharmacy and diagnostic laboratories of biosafety level (BSL) 2 and BSL 3+ with positive and negative air pressure rooms were designed and delivered to Monrovia, Liberia [31,32,33] for training and vaccine development purposes [34,35,36].

In 2020 and 2021, BCM built and deployed two Smart Pod isolation clinics at the Aldine area in unincorporated central Harris County (Texas) and Pasadena, TX, a city east of Houston and the second-largest city in Harris County, providing infrastructure [37] and educational support [38] for community-based management of COVID-19 through testing and administration of vaccines (Figure 1).

The goal of this pilot project is to validate the current clinical/operational protocols for infection control in the Smart Pod isolation clinic by obtaining air-distribution system and surface samples for SARS-CoV-2 and bacterial pathogens before, at the peak, and after a full clinical workday of patient care, which will include all standardized cleaning and disinfection standard operating procedures (SOP) in both positive and negative pressure zones of the Smart Pod [39].

Here, we first describe the features of the Smart Pod so that other mobile clinics can implement similar infection control measures. We then describe our surface sampling protocol and the laboratory procedures utilized to isolate bacteria and viral RNA. We proceed to report the bacterial and viral inoculum on all sampled surfaces over the three sampling points and discuss our results to show that the Smart Pod is a safe clinical environment for vaccine delivery. Altogether, we hope that our work improves vaccine uptake in underserved communities through increased confidence in vaccine delivery for healthcare workers and vulnerable populations.

## 2. Materials and Methods

The Smart Pod isolation clinic features an “Americans with Disabilities Act” (ADA)-compliant negative pressure isolation room (Exam Room 1) for severely symptomatic patients and two additional examination rooms (Exam Rooms 2 and 3) with enhanced positive air ventilation that can be used for vaccine administration. In its transport mode, the Smart Pod is the size of a standard 8′ × 20′ shipping container (Figure 2a). The collapsible unit expands into a near 400-square-feet, square-shaped medical facility (Figure 2b).

The negative pressure isolation room is operated by a Bag-In Bag-Out (BIBO) air purification system with a HEPA filter (Camfil, Stockholm, Sweden) that allows elimination of 99.7% of pathogens (Figure 2c–e). In accordance with the CDC recommendation [40], the negative pressure isolation room uses 12 air changes per hour (ACPH). This means that after using the isolation room for an infectious patient, medical personnel have to close the door of the empty isolation room and wait at least 35 min before re-entering the room. In Figure 2e, an electronic micromanometer demonstrates a negative value of air pressure behind the closed door of the isolation room.

The positive pressure air-distribution system is enhanced by an energy-efficient, self-cleaning needlepoint bipolar ionization (NPBI) ozone-free air-purifier (Global Plasma Solutions Inc., Charlotte, NC, USA) designed to handle up to 2400 cubic feet per minute (CFM) and a short-wavelength ultraviolet (UV-C) light air-purification system that covers approximately 50 CFM (Vidashield UV24, Medical Illumination, San Fernando, CA, USA). An individual air-duct branch is provided for the cross-contamination area and each of the exam rooms (Figure 2f). Using a velometer to measure the speed of air at each outlet of the air supply system (Figure 2g), we ensured nearly 10 ACPH in the positive pressure area, adding protection against airborne infection. The UV-C air-purifier treats 10 × 10 × 8 cubic feet volume of air that circulates within the pass-through corridor between the check-in area and Exam Rooms 2 and 3.

Antibacterial flooring, aseptic cleanroom walls, a foot pedal-controlled sink, antimicrobial disposable curtains, and bleachable surfaces (Figure 2c,d) are supplied with the goal of providing the same level of safety as a standard stationary hospital facility.

In this study, infection control was investigated via evaluation of surface and air-filter sampling for the presence of viral and bacterial pathogens as a means of evaluating infection control. Due to unprecedented circumstances during the pandemic and given the volume of patients seen (with known and unknown SARS-CoV-2 infection status), collecting patient metrics has not been feasible. In terms of provider metrics, after 11 months of daily usage, none of the healthcare workers (2 nurse practitioners, 3 medical assistants, 1 community health workers, 1 clerk—each day, 2 shifts) have tested positive for SARS-CoV-2.

In Figure 3, a diagram presents the steps imperative for the evaluation of environmental infection. Starting with the verification of regulatory requirements (domestic, national, and international) for hospitals and mobile isolation clinics, it is recommended to check on the most updated CDC guidance for surface disinfection procedures and air-exchange cycling. After this, the determination of operating hours and testing times should take place to assure sampling in (1) the clean clinical facility before operating hours, (2) presumably contaminated facility at peak hours, and (3) at the time after the disinfection procedure performed in accordance with CDC recommendations. Surface sampling spots must be pointed out to the patient and personnel traffic area, taking into account all three dimensions (floor, walls, medical and office equipment and furniture) [22,41]. It is crucial to understand the air flow distribution scheme and elements of air distribution system to select air-sampling spots [24,25]. Then, target agents ought to be determined and prioritized considering both viral and bacterial pathogens. Subsequently, testing and sampling methods should be evaluate based on their accessibility and rigor. Afterwards, data collection has to be carried out on the selected spots at the determined times. Analysis of the obtained testing results requires consideration of the materials used for construction and equipment of the evaluated mobile health clinic and compliance with accepted disinfection protocol. Lastly, limitations of the study have to be comprehended and discussed.

In the COVID-19 Smart Pod isolation clinic, surface and air-filter grille sampling was performed at 7 AM prior to opening, at 5 PM following the final patient encounter, and at 7 PM following the cleaning protocol (Figure 4).

We sampled floors, walls, door bars, exam tables, and countertops in high-traffic areas using CDC best practices [41] (Figure 5). We sampled the wall and floor in the negative pressure isolation room to verify that the isolation room would remain pathogen-free, even in cases of decompensated patients with COVID-19. The wall near the hand sanitizer was sampled to detect potential transfer of pathogens while in use by patients.

For bacterial sampling, the swab from a sterile sampling kit (Copan eSwab 480C) was wetted in the storage solution (Amies medium). For SARS-CoV-2 sampling, the swab from a sterile sampling kit (Yocon MT0301) was wetted in the storage solution (Hank’s solution). To collect both bacteria and SARS-CoV-2 samples, a 10 cm^2^ area was sampled in a back-and-forth motion, followed by two more passes at 90° to the previous pass. The swab was then placed into the sample tubes and stored at 4 °C until plating or qPCR. This protocol was repeated at each time point.

Bacterial detection samples were plated onto Methicillin-Resistant *Staphylococcus aureus* (MRSA) detection agar (Thermo Scientific, Waltham, MA, USA, Oxacillin resistance screening agar base #CM1008B with ORSAB selective supplement #SR0195), Vancomycin-Resistant Enterococcus (VRE) detection agar (Thermo Scientific Spectra™ VRE Medium #R01832), and blood agar (Thermo Scientific Remel™ Tryptic Soy Agar #R455006 with 5% defibrinated sheep’s blood #R54008). Plates (4 plates for each sample) were placed in a 37 °C incubator for 16 h. Single colonies were then counted.

We chose to measure MRSA and VRE colonies, since MRSA and VRE are common nosocomial pathogens that can cause serious infections. While the incidence of MRSA and VRE exposure in mobile clinics remains unknown, we nevertheless sought to quantify MRSA and VRE colonies in order to suggest that the infection control measures employed in the Smart Pod are robust and comparable to the infection control expectations in a large community hospital.

SARS-CoV-2 detection was performed according to the CDC EUA protocol. To extract RNA, 200 µL of each sample was extracted using Purelink Pro 96 Viral RNA/DNA Purification Kit (Invitrogen, Carlsbad, CA, USA, 12280096A). Detection of SARS-CoV-2 was accomplished by qPCR. Each sample was added to a reaction containing TaqPath 1-Step RT-qPCR Master Mix (Applied Biosystems, Waltham, MA, USA, A15299) and a primer and probe set acquired from IDT (10006770). Assembled reactions were run on an Applied Biosystems Quantstudio DX (4480299). Reverse transcription was accomplished at 25 °C for 2 min, 50 °C for 15 min, and 95 °C for 2 min. The RT-qPCR reaction cycle conditions were 95 °C for 3 s, then 55 °C for 30 s (repeated for 45 cycles).

## 3. Results

In conjunction with Harris County Precinct 2, Baylor College of Medicine has provided the mobile clinic infrastructure [26,37] and educational support [38] for community-based free COVID-19 care, available to everyone, without requiring proof of citizenship, residence, or insurance. Patient visits for COVID-19 management and follow-up care began on 3 August 2020. As of 15 July 2021, nearly 14,289 appointments for COVID-19 testing, COVID-19 management, and follow-up care have been provided to the Aldine area in central Harris County (Texas)—about 60% of served patients—and Pasadena, TX, a city in Harris County—about 40% of the served patients. Of those 14,289 patient encounters, 10,746 appointments were offered to provide COVID-19 testing alone. All other patient visits (*n* = 3543) addressed symptoms of upper respiratory illness/COVID-19 management, chronic disease management, annual physical examination, and miscellaneous, uncategorized presentations. Furthermore, more than 3000 vaccines have been administered in the Smart Pod.

In the COVID-19 Smart Pod isolation clinic, all patients that require respiratory or droplet precautions have to be examined inside the negative pressure isolation room. To increase infection control, we implemented a traffic scheme to segregate symptomatic and asymptomatic patients and created an isolation ward that would quarantine infected “mask-off” patients. Once vaccines became available, we pivoted to providing both COVID-19 vaccination in addition to COVID-19 testing.

On the sampling day, 37 patients (adults (18+)—32, pediatric—5) visited the Smart Pod by the following categories:-Visit type: sick visit—5, annual physical—1, other vaccine—1;-COVID-19: vaccine—16; test—14, positive—2, negative—11.

In Table 1, we demonstrate that SARS-CoV-2 swabs for all surfaces were negative at all time points, except for the positive control.

Additionally, except for the positive control, all surfaces were negative for MRSA and VRE colonies at all time points. Overall bacterial inoculum (measured as colonies/cm^2^) was also relatively low. We show the pattern of change in the bacterial load of sampled surfaces across three time points in Figure 6.

The highest biological contamination was detected on the floor in both positive pressure air flow and negative pressure air flow areas, while CFU data from other surfaces confirmed that the negative pressure isolation room was characterized as a cleaner environment. In particular, aseptic walls of the isolation room were endorsed to have zero bacteria.

## 4. Discussion

We have demonstrated the absence of COVID-19 contamination in the Smart Pod. Our sampling date coincided with the visit of two SARS-CoV-2-positive patients to the Smart Pod, which adds further significance to our results, as we are able to show sterile surfaces in the context of a possible occasion for SARS-CoV-2 contamination. The absence of MRSA and VRE from all samples is also notable. While absence of MRSA and VRE might not necessarily correlate with clinical outcomes, these data still show advanced infection control of the environment.

The pattern of total CFU change over the three sampling points is also informative. During the peak hours of operation (5 PM), a spike in bacterial inoculum was observed across all surfaces, yet following disinfection, the bacterial inoculum decreased. Furthermore, in the morning, prior to the start of operations (7 AM), the bacterial inoculum was similarly low, suggesting the persistence of a sterile environment overnight in the Smart Pod.

In Figure 6a, the 7 PM rise in CFU concentration on the countertop is another demonstration of the importance of the strict disinfection procedure performed by personnel. It is unclear why the CFU load rose on the countertop following disinfection. However, we note that prior to deployment of the Smart Pod, all staff were trained to ensure that high-contact surfaces remained clean. Specifically, we stressed the scrupulous sanitation of door handles, patient tables, nurses’ stations, and site of COVID-19 test storage. Such educational measures comprise an important piece of the objective of maintaining a sterile environment for COVID-19 testing and vaccine administration.

Additionally, during peak operation time (5 PM), the air purification system worked to significantly blunt the bacterial load. CFU remained relatively low during the morning, although a slight uptick was observed (Figure 6c). While the precise reason for this slight increase is unclear, we speculate that the absence of UV-C air purification during the nighttime resulted in a slightly higher bacterial inoculum during the morning. In the negative pressure isolation room (Figure 6d), there was a twofold drop in bacterial load from the air return vs. air supply; in the positive pressure area, a greater than seven-fold decrease was observed between the air return and air supply (Figure 6c). Accordingly, we show that the Smart Pod air purification system worked to decrease bacterial inoculum.

Thus, we meticulously analyze the infection control features of the COVID-19 Smart Pod isolation clinic that provided enhanced patient and personnel safety in the mobile medical facility. Altogether, the Smart Pod can offer optimal conditions for the administration of vaccines, while also dually serving as a facility to provide acute care for severely symptomatic patients.

The Smart Pod isolation clinic employs and builds upon many of the infection control measures utilized by clinical facilities during the COVID-19 pandemic. We designed the facility to ensure 6-feet distancing and unidirectional patient flow. In designing the Smart Pod, we also strived to minimize airborne transmission of SARS-CoV-2 and respiratory pathogens. Morawska et al. recommended the use of germicidal ultraviolet to combat the airborne transmission of SARS-CoV-2 [42]. Likewise, we developed a UV-C air purification system additionally coupled with an NPBI air ventilation. Antimicrobial curtains have also shown efficacy in reducing bacterial inoculum [43,44] and we employed these in the Smart Pod, along with aseptic walls and antibacterial flooring. The most significant differentiator of the Smart Pod isolation clinic with other mobile medical units is a negative pressure isolation room to contain any patients that may decompensate upon arriving in the Smart Pod.

The CDC emphasized triage to mitigate further spread of COVID-19 [45], and we have implemented a triage model to minimize COVID-19 spread. Patients can be screened outside the mobile unit, and those with symptoms of frank COVID-19 infection can be seen in the negative pressure room. Testing can be performed outside, and if possible, in a drive-through arrangement; vaccinations, however, can be carried out inside. Indeed, vaccinations can also be performed outdoors, yet in resource-limited settings where it may be difficult to maintain the vaccine cold chain, it may be logistically more practical to perform the vaccinations inside. Additionally, maintaining unilateral patient flow will also be beneficial in limiting infection spread, as person to person contact will naturally be minimized. Use of antimicrobial flooring and curtains in addition to aseptic walls will limit pathogen surface inoculum and HEPA filtering, needle-point bipolar ionization, and a UV-C air-purification system will eliminate any aerosolized particles. Last but not least, education of staff to scrupulously clean high-contact surfaces should also be a priority, as was highlighted in a number of publications [22,23,46,47].

Equipped with high-standard internet connectivity, the Smart Pod functionality can also include components of telemedicine and telementoring to provide guidance and education to frontline community healthcare workers. Telementoring, by models such as Project ECHO [48,49], allowed for highly motivating, interactive sessions on the latest best practices, which were especially critical during a pandemic with rapidly changing guidance around COVID-19 testing, management, vaccine administration. Smart Pod healthcare workers were able to implement best practices and be able to counsel patients with new information as it emerged in the literature, especially as challenges such as vaccine hesitancy arose. Specific questions and case discussions about COVID-19 management and vaccine delivery were discussed and addressed to improve the confidence of healthcare workers, increase awareness, and reduce vaccine hesitancy [38].

We hope that our findings will encourage the deployment of other mobile facilities, employing similar infection control measures to minimize the risk to patients and healthcare workers within the mobile facility. We further hope that the Smart Pod will allow for easier access and a novel approach to COVID-19 vaccine delivery, especially in underserved communities with higher rates of vaccine hesitancy [50].

Our study should be interpreted in the context of several important limitations. Due to a lack of standardization and well-defined terminology for mobile clinics (e.g., mobile vs. relocatable, portable vs. on-wheels, container vs. van, expandable vs. non-expandable), the keyword-based literature search and data comparison might be deficient. To the best of our knowledge, this work is the first comprehensive study of environmental infection control in such clinics. In addition, we recognize an important limitation of our study to be the use of surface disinfection as a proxy measure of infection control. Nonetheless, no providers tested positive during the evaluation period. Given the patient population, collecting statistics on patient infection rate is not feasible, nor is it a valid measure of the efficacy of the Smart Pod in preventing infection, as many incoming patients are positive and may have acquired infection from the outside.

In the future, we will focus on development of a Smart Pod educational platform for healthcare workers in community-based clinics, employing augmented reality (AR) and virtual reality (VR) technology and automation of disinfection solutions when it is possible.

## 5. Conclusions

In summary, we have shown that the Smart Pod is an innovative paradigm for vaccine delivery in disaster and pandemic situations that also adheres to established standards of infection control and prevention. To build confidence in community-based vaccine delivery infrastructure, we have demonstrated that well-designed mobile clinics are capable of providing a safe clinical environment with comprehensive biological decontamination for both patients and healthcare workers. We stress the necessity of regular training of personnel in disinfection protocols and a high standard of services in community-based clinical sites in resource-constrained settings.

## 6. Patents

Patent WO2016077466A1 “Mobile Clinics”. Nos. 62/078,924, filed 12 November 2014, and 62/237,138, filed 5 October 2015.

## Figures and Tables

**Figure 1 vaccines-09-01362-f001:**
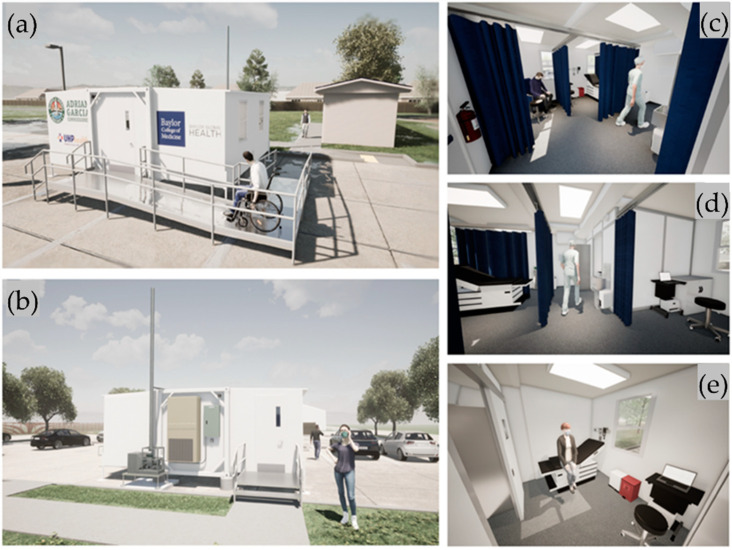
Illustration of a 3D-rendered model of the COVID-19 Smart Pod isolation clinic. (**a**,**b**) Exterior of the portable, expandable container unit: entrance and exit, respectively; (**c**,**d**) Interior: patient examination area (Exam Rooms 2 and 3) separated by disposable antimicrobial curtains and check-in area (positive pressure zone); (**e**) Negative pressure isolation room (Exam Room 1).

**Figure 2 vaccines-09-01362-f002:**
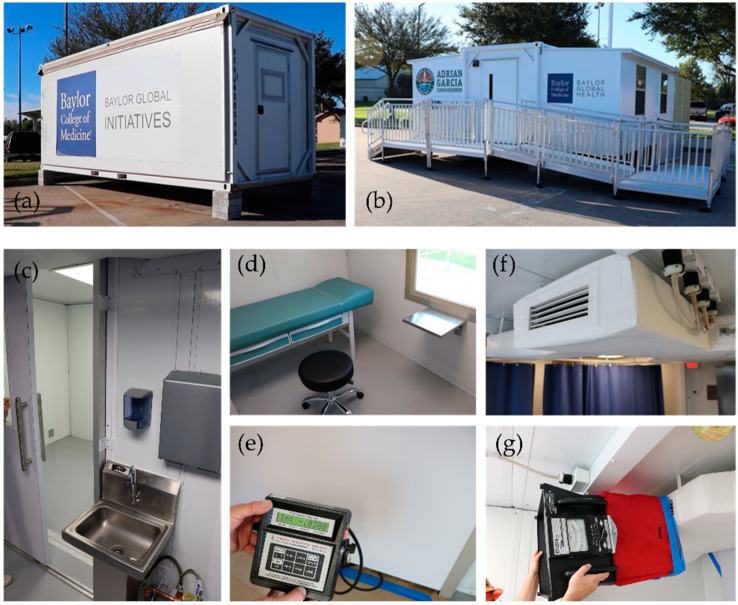
Photographs of the COVID-19 Smart Pod isolation clinic. (**a**) Folded unit in its transport mode; (**b**) Expanded container-based clinic with an ADA-compliant ramp; (**c**) Entrance to the isolation room; (**d**) Interior in the isolation room features bleachable surfaces; (**e**) A micromanometer shows measurements of negative pressure in the isolation room; (**f**) One of the air supply outlets of the air distribution system; (**g**) Positive pressure evaluation in one of the air supply outlets using a velometer.

**Figure 3 vaccines-09-01362-f003:**
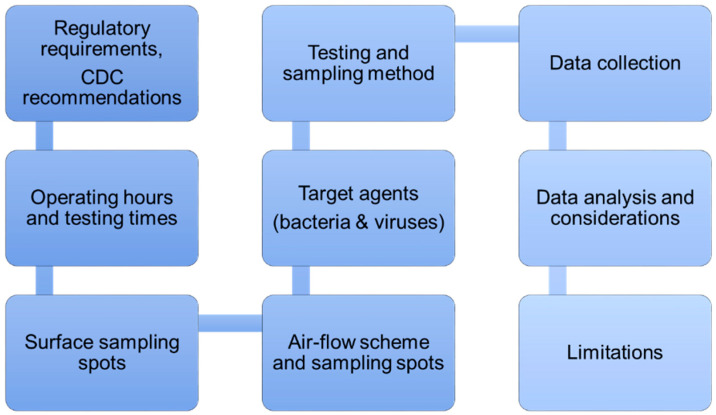
Development of an environmental infection evaluation plan. Schematic guideline for development of an evaluation procedure to assess biological decontamination in a mobile container medical unit.

**Figure 4 vaccines-09-01362-f004:**
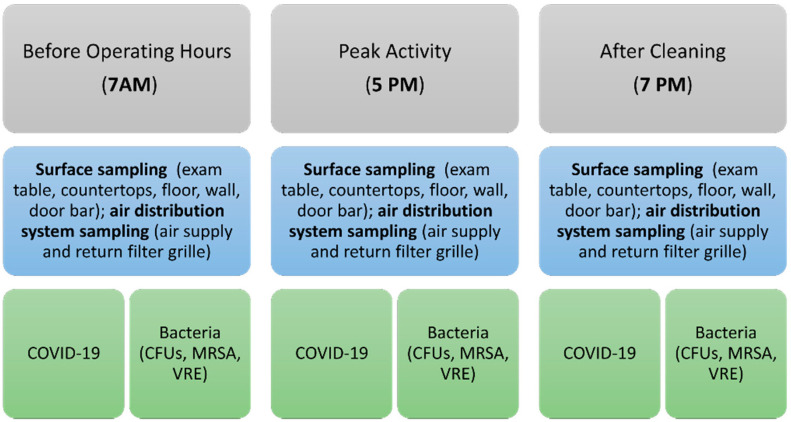
Diagram of the experimental procedure. Abbreviations: CFUs: colony forming units; COVID-19: coronavirus disease 2019; MRSA: methicillin-resistant *Staphylococcus aureus*; VRE: vancomycin-resistant enterococcus.

**Figure 5 vaccines-09-01362-f005:**
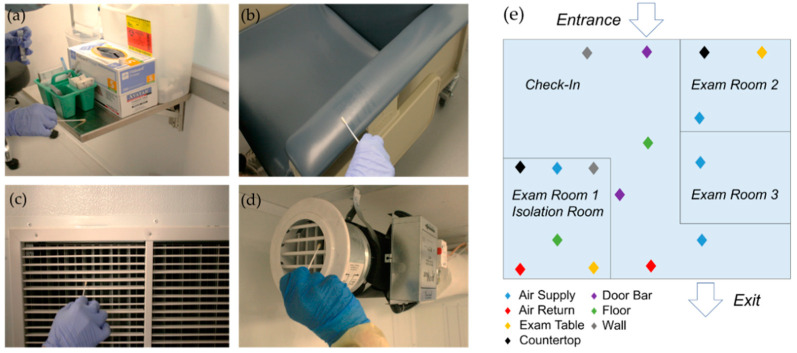
Sampling surfaces and air-distribution system. (**a**) Countertops; (**b**) Exam table/reclining chair arm; (**c**) Air return filter grille in the area of positive pressure air flow; (**d**) Air supply in the area of negative pressure air flow (isolation room); (**e**) Schematic of the mobile isolation clinic and labeled sampling areas.

**Figure 6 vaccines-09-01362-f006:**
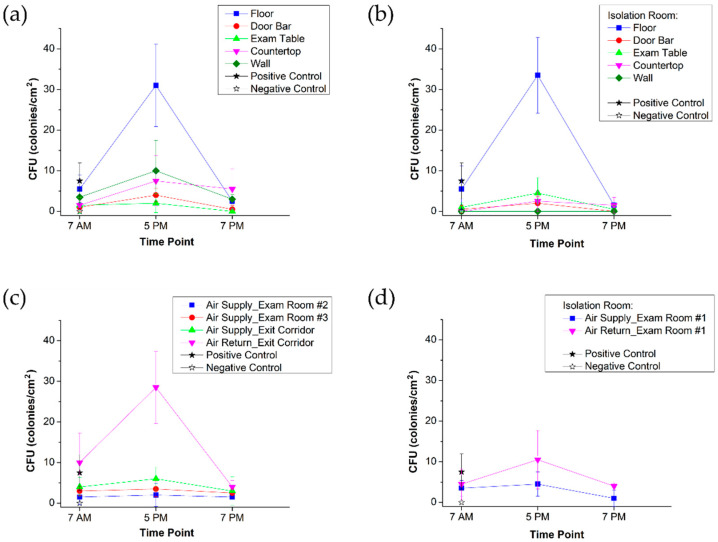
Bacteria detection on the selected surfaces and air distribution system. The mean values: (**a**) CFUs collected on surfaces in the positive pressure area across three time points. (**b**) CFUs collected on surfaces in the negative pressure isolation room across three time points. (**c**) CFUs observed on air-filter grilles at Air Supply and Air Return outlets in positive pressure area. (**d**) CFUs observed on Air Supply and Air Return filter grilles in negative pressure isolation room.

**Table 1 vaccines-09-01362-t001:** Experimental results. Bacteria and virus detection on surface samples and air distribution system in the Smart Pod isolation clinic.

Sample	Time	COVID-19 Detection (N1 Ct)	General Bacterial Culture * (CFUs/cm^2^)	MRSA (CFUs/cm^2^)	VRE (CFUs/cm^2^)
Positive Control		33.8	8	11	6
Negative Control		N.D.	0	0	0
Surface Samples (10 sampling spots)	7 AM	N.D.	2	0	0
	5 PM	N.D.	4	0	0
	7 PM	N.D.	0	0	0
Air Distribution System (6 sampling spots)	7 AM	N.D.	3	0	0
	5 PM	N.D.	6	0	0
	7 PM	N.D.	2	0	0

*: Median value of measurements obtained in four plates. N.D.: not detected.

## Data Availability

The data that support the findings of this study are available from the corresponding authors upon reasonable request.

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
