# Peer review of "A Community-Based Management of COVID-19 in a Mobile Container Unit"

_vaccines, 2021, doi:10.3390/vaccines9111362_

Round 1
Reviewer 1 Report
Unfortunately, I cannot see the responses the authors gave to my previous observations. I confirm my impression: the article completely lacks comparison with similar experiences and therefore looks like an advertisement for a mobile medium, rather than a scientific contribution. Readers learn that there is a mobile medium, that it is safe, but they know nothing about other ways of dealing with the problem. I guess it is not the first time that someone has proposed the vaccine delivery operation and therefore there should be other experiences in the field. The Discussion is completely devoid of references.
Author Response
We would like to thank the Reviewer for critical reading of our manuscript and giving us an opportunity to improve our manuscript. A mobile container unit solution is proposed as an augmentation to conventional healthcare facilities and temporary mass vaccination centers that cannot address accessibility needs of underprivileged and vulnerable population in remote area. While we are not aware about any studies that validate the infection control measures and biological decontamination of a mobile facility, we have added a comparison of typical challenges for mobile health clinics in the Introduction (page 2, lines 53-65) and the Discussion (page 10, lines 359-364) and the according reference to our manuscript. We hope that readers could learn more about mobile medium through these comments.
Reviewer 2 Report
Estimated Authors of the paper "A Community-Based Management of COVID-19 in a Mobile Container Unit"
I've read with great interest the present report, accurately depicting this new item available to counter the ongoing SARS-CoV-2 pandemic, i.e. a mobile vaccination center.
During the initial stages of the vaccination campaign, mass vaccination centers (see for example https://pubmed.ncbi.nlm.nih.gov/34205891/) have been extensively employed in order to obtain high vaccination rates in relatively short time. At the moment, having reached a significant share of the general population, and without clear recommendations for pediatric-age vaccination, most of mass vaccination centers have been shutten down. On the contrary, the availability of smaller units, quick to deploy, that the healthcare authorities can employ even in complicated settings (e.g. in rural area, but also in centers that have been recently hitten by natural disasters) may allow to sustain the vaccination campaign, possibly improving the vaccination rates in population groups that have been only marginally vaccinated because of the unfavorable accessibility to the vaccination centers.
In other words, I think that this paper may be very interesting for all public health authorities.
Unfortunately, reporting on a new technology (without performing a true HTA) may be quite difficult, and the Authors may face the risk of being unable to grasp the attention of their readers. In this case, Authors, have been conscious of the potential limits of their approach, and were unable to perform a true HTA because of their settings ("Due to unprecedented circumstances during the pandemic and given the volume of patients seen (with known and unknown SARS-CoV-2 infection status), collecting patient metrics has not been feasible"), but - on the other hand, their have provided to the readers accurate data about all measurable factors they were able to collect, and also detailed information that may allow some kind of reproducibility of this intervention (e.g. designing a similar Mobile Unit).
For the aforementioned reasons, I'm advocating the acceptance of this paper as it is.
Author Response
We greatly appreciate the Reviewer’s comments on importance of the proposed vaccine delivery method. We will elaborate on patient metrics and other measurable factors with expansion of our program.
Reviewer 3 Report
I found the topic of the paper interesting. With all these, there are new additions in the paper which are not properly justified in the paper either by studies or by referring scientific works - e.g. in the abstract the authors state that "Mobile clinics hold the promise of ameliorating such inequities, although there is paucity of studies that validate the environmental infection control in such facilities, which is an important contributing factor to vaccine hesitancy" - this affirmation is not supported by any data/reference in the Introduction. Please add some information in the Introduction to support this point of view. The literature related to COVID-19 vaccination hesitancy is vast - there are studies based on questionnaires and sentiment analysis - I believe that there are also studies which support the authors' point of view. Please add them in order to support your point of view.
Please add a roadmap of the study (in words) at the end of section 1 for better explaining the organization of the paper.
Please add a scheme in the Material and Methods section in which you highlight the steps needed to be taken in order to implement the proposed approach even to other similar mobile container units.
As for the Results section, I believe that the authors should compare their approach with the current way of using the mobile container unit in order to better highlight the proposed approach. Please add also some indicators in order to prove the efficiency of the proposed approach (similar to the ones you have used in Table 1).
Please do not use acronyms in the text before explaining them (e.g. in the abstract CFU, MRSA, VRE).
Please discuss the limitations of the study and how it can be generalized to similar situations.
Author Response
Please add some information in the Introduction to support this point of view. The literature related to COVID-19 vaccination hesitancy is vast - there are studies based on questionnaires and sentiment analysis - I believe that there are also studies which support the authors' point of view. Please add them in order to support your point of view.
We have added four references in paragraph two of the Introduction which elucidate the contribution of both racial and economic disparities to vaccine hesitancy through a common denominator of limited accessibility to vaccinations (page 1-2, lines 42-50).
Please add a roadmap of the study (in words) at the end of section 1 for better explaining the organization of the paper.
We have added a paragraph at the end of our Introduction section to better explain the organization of our manuscript (page 3-4, lines 112-119)
Please add a scheme in the Material and Methods section in which you highlight the steps needed to be taken in order to implement the proposed approach even to other similar mobile container units.
We have added Figure 3 to exemplify the development of an environmental infection evaluation plan and a paragraph to explain the diagram (page 5, lines 164-181).
As for the Results section, I believe that the authors should compare their approach with the current way of using the mobile container unit in order to better highlight the proposed approach. Please add also some indicators in order to prove the efficiency of the proposed approach (similar to the ones you have used in Table 1).
We have discussed typical advantages and disadvantages of mobile health clinics in the Introduction (page 2, line 53-62). Moreover, we will continue our study and collect more information on metrics of acceptability of the mobile container units in future.
Please do not use acronyms in the text before explaining them (e.g. in the abstract CFU, MRSA, VRE).
Thank you for your comments. We have removed acronyms from the Abstract (page 1, line 22-23) and added a few expansions of abbreviations (page 1, line 39; page 4, line 122).
Please discuss the limitations of the study and how it can be generalized to similar situations.
We have discussed more limitations in the Discussion (please see page 10, lines 354-359).
Round 2
Reviewer 1 Report
The manuscript has had some improvements since the first version
Reviewer 3 Report
Thank you for the revised version. I have no further comments.
This manuscript is a resubmission of an earlier submission. The following is a list of the peer review reports and author responses from that submission.
Round 1
Reviewer 1 Report
This article illustrates the advantages of a mobile vehicle designed to operate in situations of particular logistical difficulty. The authors evaluated the efficiency of the infection control and biological decontamination system. The samplings were all done in one day.
- The authors report the test results and in the same paragraph discuss the results. it would be preferable to have two different chapters for results and discussion.
- Before discussing the methods of confinement of the SARS-CoV-2 virus, the authors could have made some mention of its spread in closed environments and in air conditioning systems [Chirico F, Sacco A, Bragazzi NL, Magnavita N. Can Air-Conditioning Systems Contribute to the Spread of SARS/MERS/COVID-19 Infection? Insights from a Rapid Review of the Literature. Int J Environ Res Public Health. 2020 Aug 20;17(17):6052. doi: 10.3390/ijerph17176052]
- In the article, there are no indications about other mobile means with which to make a comparison.
- Overall, given the limited importance of the part dedicated to the dosage of biological contamination, the text is very similar to an advertising brochure. Some indications, such as line 90 in materials and methods (“which can be shipped by sea, land, or air”), are taken from the commercial advertisements of the product and are totally out of place in this context. Authors should give the text the style of a scientific paper.
Reviewer 2 Report
This is an exciting piece of work. However, I wonder, though, whether it has much scope in the journal vaccines. Therefore, I reject it since it does not have much scope in vaccines.
You might want to consider submitting it to another journal.